# Preoperative Assessment of Language Dominance through Combined Resting-State and Task-Based Functional Magnetic Resonance Imaging

**DOI:** 10.3390/jpm11121342

**Published:** 2021-12-09

**Authors:** Christian Ott, Katharina Rosengarth, Christian Doenitz, Julius Hoehne, Christina Wendl, Frank Dodoo-Schittko, Elmar Lang, Nils Ole Schmidt, Markus Goldhacker

**Affiliations:** 1Department of Neurosurgery, University Regensburg Medical Center, 93053 Regensburg, Germany; Katharina.Rosengarth@ukr.de (K.R.); Christian.Doenitz@ukr.de (C.D.); Julius.Hoehne@ukr.de (J.H.); Nils-Ole.Schmidt@ukr.de (N.O.S.); Markus.Goldhacker@ukr.de (M.G.); 2Department of Radiology, University Regensburg Medical Center, 93053 Regensburg, Germany; Christina.Wendl@ukr.de; 3Institute of Epidemiology, University of Regensburg, 93053 Regensburg, Germany; Frank.Dodoo-Schittko@klinik.uni-regensburg.de; 4Computational Intelligence and Machine Learning Group, Institute of Biophysics, University of Regensburg, 93053 Regensburg, Germany; elmar.lang@biologie.uni-regensburg.de

**Keywords:** resting-state fMRI, task-based fMRI, brain mapping, language assessment, data-driven analysis

## Abstract

Brain lesions in language-related cortical areas remain a challenge in the clinical routine. In recent years, the resting-state fMRI (RS-fMRI) was shown to be a feasible method for preoperative language assessment. The aim of this study was to examine whether language-related resting-state components, which have been obtained using a data-driven independent-component-based identification algorithm, can be supportive in determining language dominance in the left or right hemisphere. Twenty patients suffering from brain lesions close to supposed language-relevant cortical areas were included. RS-fMRI and task-based (TB-fMRI) were performed for the purpose of preoperative language assessment. TB-fMRI included a verb generation task with an appropriate control condition (a syllable switching task) to decompose language-critical and language-supportive processes. Subsequently, the best fitting ICA component for the resting-state language network (RSLN) referential to general linear models (GLMs) of the TB-fMRI (including models with and without linguistic control conditions) was identified using an algorithm based on the Dice index. Thereby, the RSLNs associated with GLMs using a linguistic control condition led to significantly higher laterality indices than GLM baseline contrasts. LIs derived from GLM contrasts with and without control conditions alone did not differ significantly. In general, the results suggest that determining language dominance in the human brain is feasible both with TB-fMRI and RS-fMRI, and in particular, the combination of both approaches yields a higher specificity in preoperative language assessment. Moreover, we can conclude that the choice of the language mapping paradigm is crucial for the mentioned benefits.

## 1. Introduction

Tumors in language-related areas remain challenging in neurosurgery. Brain-mapping measures are a balancing act between maximal tumor resection and enhancing patient survival [1]. The preservation of eloquent areas improves health-related quality of life [2] by reducing the risk of neurological impairment [1,3,4]. Because of interindividual anatomic variability, preoperative assessment and intraoperative cortical mapping are often required to optimize clinical outcome [4,5,6]. This is especially important in presurgical language mapping and identification of language laterality.

It is well established that critical language functions are lateralized in one hemisphere [7]. Most healthy subjects show an unambiguous hemispheric dominance according to language processing in the left hemisphere. Only approximately 10% of healthy subjects show a right hemispheric language representation [8]. Mixed language dominance might occur in rare cases following infantile brain lesions presumably triggering neuroplastic processes, or in psychiatric disorders such as schizophrenia [9], autism [10], or dyslexia [11].

The gold standard for identification of language-related brain areas is direct cortical stimulation (DCS) during awake surgery [4,5,6,12], enabling a relatively precise and reliable determination of language-related areas. This method is invasive by definition and, thus, a preoperative and noninvasive identification for surgery planning is not feasible using direct cortical stimulation.

One of the most prominent presurgical methods of language lateralization assessment is task-based magnetic resonance imaging. TB-fMRI has been used to overcome the limitations of DCS in the preoperative assessment [6]. The primary advantage of this technique is its noninvasive nature and the possibility to identify eloquent areas preoperatively. Thus, this method may influence the surgical strategy, not only by showing a safer approach to the tumor but even by changing the type of surgical procedure.

TB-fMRI has some drawbacks limiting its use for functional assessment. Firstly, patients have to be able to perform the task. Patients with a tumor close to eloquent areas may be less able to cooperate because of neurological impairment [13]. Secondly, patients must be awake to perform the task. Because sedatives cannot be administered, the use of TB-fMRI is limited, for instance, in pediatric or claustrophobic patients [3]. Further drawbacks are the variability of results using different language tasks [13], motion artifacts [14], and low signal-to-noise ratio [15]. In comparison to DCS, the specificity and sensitivity of TB-fMRI are mediocre and—according to several studies—highly variable [16]. Reasons might be inefficient experimental designs or different language tasks and methods of analysis [17]. In the case of multiple functional analyses, TB-fMRI is time-consuming. Furthermore, TB-fMRI results of the underlying individual anatomy may be challenging to interpret because of the inseparability of simultaneously examined tasks, for instance, by activation of the visual cortex, attention networks, and working memory during reading, when patients perform language tasks [12,13,18]. Misinterpretations may result in a more ambiguous rate of lateralization indices, which could—in theory—lead to incorrect decisions regarding the surgical strategy.

In recent years, a new technique of fMRI—termed resting-state magnetic resonance imaging (RS-fMRI)—has been proposed as an imaging method for preoperative localization of eloquent areas [19]. The primary advantage of RS-fMRI is the complete abandonment of any task because no patient participation is required [19]. Several studies have shown that RS-fMRI examination is, in fact, independent of a patient’s level of consciousness. Stable results can even be yielded during sleep [20,21,22] and anesthesia [23,24,25,26]. Because RS-fMRI can be used irrespective of the sedation and cognitive status of the patient [3,19], this method seems to be substantially more widely applicable [3]. RS-fMRI examines the endogenous brain activity and is based on low-frequency fluctuations in the BOLD signal [27]. Different methods for analysis are available. For analyzing language networks in tumor patients, a seed-based approach seems to be impractical because of the high interindividual anatomic variability of cortical language representations, and language lateralization. The need to select seed regions represents a considerable limitation of a seed-based analysis [28]. Few authors have claimed a data-driven approach using independent component analysis (ICA) as a proper alternative for the analysis of language networks [3,13].

Some authors have described a more considerable specificity of RSLN than of task-based analysis. Task-based activation is not restricted to critical language regions because it contains regions involved in visual processes, attention, working memory, and others. Thus, the resting-state analysis seems to extract language regions more selectively than task-based methods and allows the identification of the critical brain regions for language while excluding non-language-related processes [13].

According to Dodoo-Schittko et al., language lateralization may be emphasized by applying various language paradigms, including cognitive high-level control conditions, to separate language-critical processes from language-supportive processes in TB-fMRI [17]. The application of various language paradigms may also lead to more unambiguous results for cortical language representations, thereby affecting preoperative planning [17]. Though, a syllable switching task may serve as a high-level cognitive control condition for word generation tasks to reveal pure language-critical processes [17].

TB-fMRI may be less specific due to the involvement of other cortical regions during the performance of tasks, thereby leading to less specific results and more ambiguous lateralization indices. A method to partially overcome this limitation in TB-fMRI could be the additional use of the above-mentioned control condition. On the other hand, identifying language-relevant cortical representations solely through RS-fMRI underlies some drawbacks because of language lateralization and interindividual anatomic variability. Our study aimed to examine whether a method using an identification algorithm for the RSLN based on the individual TB-fMRI results could lead to significant changes in language lateralization measurements. We used the control condition mentioned above to obtain even more specific language-relevant cortical representations during TB-fMRI.

## 2. Materials and Methods

The local ethics committee approved this study: Ethikkommission an der Universität Regensburg (Universität Regensburg, Ethikkommission, 93040 Regensburg, Germany). Twenty patients suffering from lesions close to the language-relevant cortical areas were included (12 men and 8 women). fMRI was conducted for preoperative language assessment. The mean patient age was 46.97 ± 13.13 years. MRI scans were conducted with a Siemens 3 Tesla MRI scanner (Magnetom Allegra, Siemens, Erlangen, Germany, *n* = 13) and a Siemens 3 Tesla whole-body imaging system (Magnetom Skyra, Siemens, Erlangen, Germany, *n* = 7). For further details, see Table 1.

TB-fMRI and RS-fMRI data were collected at the Center of Neuroradiology of the community hospital in Regensburg and the Institute of Radiology at the University Medical Center Regensburg between 2013 and 2017. At the community hospital, images were obtained with a Siemens 3 Tesla MRI scanner (Magnetom Allegra, Siemens, Erlangen, Germany) with a 1-channel head coil system. fMRI imaging parameters applied for functional T2∗-images were TR = 2000 ms, TE = 30 ms, flip-angle alpha = 90°, field-of-view (FoV) = 192 × 192 mm2, matrix = 3 × 3 × 3 mm3 voxel, and 34 axial slices. Slice acquisition occurred in interleaved order. Additionally, a structural T1-weighted image (TR = 2300 ms, TE = 2.91 ms, flip-angle alpha = 9°, FoV = 256 × 256 mm2) of 160 axial slices was recorded that measured 1 mm in thickness and had a voxel size of 1 × 1 × 1 mm3. fMRI data collected at the University Medical Center of Regensburg were acquired with a Siemens 3 Tesla whole-body imaging system (Magnetom Skyra, Siemens, Erlangen, Germany) with a 32-channel phased-array head coil. A T2∗-weighted gradient EPI sequence was used with the following acquisition parameters: TR = 2000 ms, TE = 35 ms, flip-angle alpha = 90°, FoV = 192 × 192 mm2, matrix = 2.3 × 2.3 × 2.3 mm3 voxel. A total of 31 axial slices were acquired in interleaved order. The imaging system also provided a structural T1-weighted image (TR = 1919 ms, TE = 3.67 ms, flip-angle alpha = 9°, FoV = 256 × 256 mm2) of 160 axial slices with a thickness of 1 mm and a voxel size of 1 × 1 × 1 mm3. The same imaging parameters were applied for TB-fMRI and RS-fMRI.

During TB-fMRI sessions, patients had to perform semantic generation paradigms, as previously described by Dodoo-Schittko et al. [17]. During the tasks, patients were trained to sub-vocally produce either a semantically related verb according to a noun or an antonym associated with an adjective. A syllable switching task served as the linguistic control condition for both generation tasks to separate language-critical processes from language-supporting processes [17]. In a block design, the sequence–word generation–fixation–control condition was repeated 10 times for each paradigm. Stimuli were presented visually.

RS-fMRI was conducted with eyes open, fixating a black cross on a grey screen. Patients were instructed to relax, and let their mind wander but not fall asleep. Depending on patient ability, between 299 and 450 (TR = 2) images were recorded (see Table 1).

Both RS-fMRI and task-related data were preprocessed using SPM12 (statistical parametric mapping), DPARSF (Data Processing Assistant for Resting-State fMRI) [29], and in-house MATLAB scripts. After slice time correction, functional images were realigned to a mean image followed by the registration of functional and structural images. Then, the structural image was segmented into gray matter, white matter, and cerebrospinal fluid. Afterward, normalization to MNI space (Montreal Neurological Institute, Canada) [30] was applied to all images, including segments followed by smoothing of functional images with a kernel of FWHM of 6 × 6 × 6 mm³.

RS-fMRI data were subjected to temporal voxel-wise detrending to remove trends up to cubic order. Then, covariates were regressed out to account for artefactual noise. Besides motion regressors, additional covariates were constructed from the white matter and cerebrospinal fluid mask of each patient. These masks were constructed by thresholding the corresponding segmented images (white matter mask included only voxels with a probability of >99% of belonging to white matter, while the probability threshold for the cerebrospinal fluid mask was >90%). To account for further motion artifact removal, the functional images were subjected to motion scrubbing using a summary measure for motion and to thresholding at 0.5 [31]. Each flagged image was removed, and the adjacent images were used to interpolate the estimated voxel time courses of the removed images using spline interpolation. The data were bandpass-filtered with a high-frequency cut-off of 0.15 Hz and a low-frequency cut-off of 0.01 Hz to remove known physiological noise sources and scanner drifts [32,33].

ICA analysis was performed using the GIFT toolbox (http://mialab.mrn.org/software/gift/index.html (accessed on 12 December 2020)). First, we estimated the data-inherent number of ICs by using the MDL criterion (Minimum Description Length) [34], which resulted in *n* = 173.5 ± 59.6 data-inherent ICs on average. To explore a sufficiently large space of ICs, we calculated ICA runs ranging from $m=n_{round}-80\leq n_{round}\leq to\;m=n_{round}+50,$ in which $n_{round}$ is the estimated number of data-inherent ICs rounded to the next decade. Each ICA run consisted of the following steps: After removing the mean of the image at each time point, *m* principal components were extracted by employing PCA (principal component analysis). Then, the ICASSO procedure [35] was applied to calculate the Infomax algorithm [36] ten times to extract the best estimates of ICs. For standardization purposes and to apply a meaningful threshold, the resulting components were z-scored.

The design matrix was identical for both language tasks. The experimental and control conditions were modeled as separate regressors by convoluting each paradigm time-course with the canonical HRF (hemodynamic response function) of SPM12. Additionally, the motion regressors were included in the design matrix as six separate regressors. The fixation period was not explicitly defined as a regressor and served as an implicit baseline. For the following analyses, two contrasts were calculated: experimental condition > control condition and experimental condition > baseline.

To select the best independent components (IC) from the large IC space of each patient, we compared all ICs from RS-fMRI of each patient with the corresponding GLM contrasts. To achieve this, we employed the Dice coefficient [13], a measure of overlapping the two sets *X* and *Y*. Set *X* was the thresholded contrast image from the GLM analysis and *Y* the thresholded IC from the ICA. For the GLM contrast images, a threshold of T = 4 was chosen and for the ICs T = 2. The Dice coefficient *D* is defined as:$$D=\frac{2\left|X\cap Y\right|}{\left|X\right|+\left|Y\right|}$$

To find the best IC, the Dice coefficient was evaluated for each IC on a single subject basis. This evaluation was performed for both contrast images resulting from the GLM analysis. For each patient, this evaluation yielded 4 ICs: the best IC from antonym or verb generation with and without control condition. The occipital lobe was masked for these calculations to remove the influence of visual cortex activations.

To assess language laterality, we calculated the laterality index on the GLM contrasts and the selected ICs from RS-fMRI using the laterality toolbox from SPM12 [37] using the bootstrapping approach for calculating a laterality measure. From the bootstrapped samples, the weighted mean laterality index (WMLI) was calculated, representing the extent of lateralization tendency in the applied ROIs (regions of interest) [38]. We constructed language-specific ROIs utilizing the WFU-Pickatlas (Wake Forest University) [39,40] including opercular, triangular, and orbital part of the inferior frontal gyrus in both hemispheres. Activations in the midline (range in the x-direction: ±10 mm) were disregarded in our laterality calculations. Since we wanted to decide whether the recently described language-inducing paradigm in combination with RS-fMRI had an advantage over conventional techniques, we used the absolute value of this WMLI from each subject to represent a measure for unambiguousness in the decision process on laterality. Testing was carried out with the Wilcoxon rank sum test.

## 3. Results

Results of the verb generation and antonym task are condensed in Figure 1 and Figure 2, which show second-level group statistics for both GLM and RS-fMRI analysis. Group-level statistics were conducted using the second-level function of SPM (this threshold was also used for extracting the information displayed in Table 2). To achieve higher specificity in the selected ICs, we applied a more conservative threshold (T = 4) when comparing the template images with all ICs resulting from ICA. The more liberal threshold (T = 3) was used for group-level images and the cluster information table. The information in Table 2 was generated with a cluster threshold of 100 voxels.

### 3.1. GLM Results

The comparison between antonym and verb generation tasks and baseline showed enhanced activation in a huge bilateral but pronounced left-hemispheric network of areas associated with visual language processing including cortical and subcortical areas of the frontal, temporal, parietal, and occipital lobe (for details, see Figure 1 and Figure 2, and Table 2). In contrast, GLM comparison between antonym and verb generation tasks and linguistic control condition showed a more specific lateralized language-associated activation pattern, mainly on the left side due to reduced, supportive language activation, for instance, early visual processing or motor planning processes. (T = 3; for details see Figure 1 and Figure 2, and Table 2).

### 3.2. RS-fMRI Results

According to the verb-generation task compared to baseline, the best matching RS-fMRI component showed bilateral activation of the inferior and superior parietal lobe, the right angular gyrus, the left middle occipital gyrus, and the left precentral and inferior frontal gyrus. In contrast, the RS-fMRI component that best matched the contrast verb-generation versus syllable-switching showed left-lateralized activation of a more language-specific pattern in the inferior and frontal gyrus, the supplementary motor area, the middle temporal gyrus, the basal ganglia, the inferior parietal lobe, and the supramarginal and angular gyrus (for details see Figure 1, Table 2).

The best matching RS-fMRI component, according to the comparison between the antonym task and baseline, showed a language-activation pattern only in the left hemispheric regions, including the inferior and precentral gyrus, the superior occipital and middle occipital lobe, and the superior parietal lobe. The RS-fMRI component that best matched the contrast antonym generation versus syllable switching also resulted in a solely left-hemispheric but more expanding activation pattern than the former baseline contrast in the inferior frontal and middle gyrus, the supplementary motor area, the middle temporal gyrus, the basal ganglia, the inferior parietal lobe, and the supramarginal and angular gyrus (T = 3; for details, see Figure 2 and Table 2).

### 3.3. Laterality Index

Figure 3 and Figure 4 show the results of laterality index calculations employing the weighted mean of the bootstrap samples as a measure of laterality. Figure 3 shows the WMLI on a single subject level for a more detailed qualitative inspection. The panels of Figure 4 include a summary of the analyses resulting from the GLM calculations and the RS-fMRI results. Each panel also shows the WMLI of the images using the task with and without control condition. The GLM results slightly differ in the median and standard deviations between images generated with and without control condition. This difference is confirmed by employing Wilcoxon rank sum tests that yielded a non-significant result for both panels (*p* > 0.47). The panels representing the RS-fMRI results show a different pattern. Using the paradigm with the control condition, the verb and antonym generation tasks yielded larger median and lower standard deviation values of the weighted mean measure. This result was also reflected in the significant Wilcoxon rank sum tests (verb: *p* < 0.0001; antonym: *p* < 0.007).

### 3.4. Dice Index

The mean and standard deviation of the Dice index correlations of the best-fitting ICs to the TB-fMRI results of verb and antonym generation with and without the linguistic control condition (verb = verb generation; anto = antonym generation; w/o LCC = without linguistic control condition; with LCC = with linguistic control condition) are represented in Table 3.

## 4. Discussion

The results of this study suggest that the combination of TB-fMRI and RS-fMRI data may not only increase presurgical identification of language-critical areas by means of specificity but also improve language laterality predictions.

First, in line with a recent study by Branco et al., we found that the application of RS-fMRI in patients with brain tumors shows language-related areas comparable to specific task fMRIs [13]. Activation was increased in a bilateral but strongly pronounced left-hemispheric network of areas associated with visual language processing, including cortical and subcortical areas of the frontal, temporal, and parietal lobe. Comparing RS-fMRI results using GLM contrasts with a linguistic control condition to those without (baseline) shows a qualitatively improved representation of language-associated areas by means of an increased specificity activation that is more pronounced in the brain areas critical for language processing. The application of a syllable switching task as a useful linguistic control condition that is effective in differentiation between language-critical and language-supportive activation was described for a cohort of healthy subjects by Dodoo-Schittko et al. (2012) [17].

Second, our study population was investigated with regard to language laterality by means of the WMLI. The use of GLM contrasts applying a linguistic control condition for selecting language-related ICs resulted in a significantly more unambiguous and higher language laterality index than without the application of a linguistic control condition. In contrast, the comparison of language laterality indices between the two GLM contrast conditions per se did not yield any significant results. The more focused and more specific activity distribution in GLM contrast with the control condition significantly increased laterality pronunciation in the RS-fMRI result, which is also reflected in the higher and more focused WMLI with the control condition than the more dispersed distribution without the control condition (compare box plots of Figure 4, bottom row). In detail, the WMLI results on the single-subject level are consistent. The GLM results for both verb and antonym generation tasks result in comparable—and, in particular, same signed—WMLIs for most patients. Dodoo-Schittko et al. assumed in 2012 that, in the case of TB-fMRI, the application of a linguistic control condition such as syllable switching may predict language laterality more unambiguously by means of higher LIs in healthy controls. This assumption could not be supported by the data of this study on the level of the GLM alone. Only the combination of RS-fMRI and TB-fMRI yielded significantly higher and more unambiguous language laterality indices due to the application of a linguistic control task. One reason why the GLM alone did not benefit from linguistic control conditions may also be the considerable heterogeneity of the patient group according to differences in brain tumor location, volume, or dignity.

Analysis solely based on RS-fMRI using predefined templates seems to be feasible. However, the results should be considered with care because the question of to what extent brain regions that are functionally interconnected are indeed active during task execution remains unanswered [13]. Additionally, using a canonical template has the drawback that it contains the lateralization information of, for instance, a healthy cohort already and, thus, could lead to problems if you arrange any lateralization examination on such a template in an individual patient. Our approach is to circumvent this limitation of a template-based analysis by using the individual TB-fMRI as an individual template for the individual patient. We aimed to enhance the preoperative assessment of language networks based on TB-fMRI by supplementing with resting-state data.

There are some limitations due to the heterogeneity of the study sample because patients vary according to tumor localization and tumor type. One patient with a neurovascular disease was included. First of all, there is literature that shows that brain tumor location affects the local distribution of language-associated activation by means of a higher bilateral activation pattern in patients with inferior frontal tumors compared to patients with tumors in temporoparietal areas [41]. Another recent study by Shaw et al. [42] reports that tumor-induced basal ganglia infiltration is associated with co-dominant language representation. Secondly, different grading of the tumor might also influence language lateralization and language laterality. In this study, most of the patients suffered from high- and low-grade glial tumors. Those tumors come along with changes in neurovascular coupling, which is tied to neural activity and, therefore, influences the BOLD signal [43,44,45]. A further limitation of this study might be the carrying out of the fMRIs on two different MRI scanners, with different head-coil systems and bore size. Otherwise, magnetic field strength and TR were constant. A recent multicenter RS-fMRI study reports less statistical power but still valid data when combining data, including different MRI-scanners [46]. In our study, both MR-scanners were 3T scanners, and the TR was kept constant, which was not the case in the latter study. A further limitation might be that our patients have not been balanced across the two different scanners and that there might be potential differences in brain coverage.

Unfortunately, we are not able to reveal information on the sensitivity, specificity, and positive or negative predictive values of our method. This could be the subject of further examination.

## 5. Conclusions

Methods for identifying resting-state language networks solely based on RS-fMRI may become increasingly important. Further development of established and new techniques may lead to improved approaches to identifying resting-state language networks in the future. On the one hand, RS-fMRI could complement task-based preoperative language assessment, and on the other hand, new analysis methods may completely replace task-based assessment in the future. The advantages of RS-fMRI are the abandonment of any task performance, facilitating a wider usage of the technique in a broader spectrum of patients, and the possibility of a post hoc analysis of a wide range of networks that can be simultaneously detected in one single time-saving MRI examination.

A data-driven technique that is mostly independent of a priori assumptions seems to be useful to obtain comparable results and to eliminate errors in the analysis, leading to extensive automation and operational availability for any patient in future routine clinical practice.

## Figures and Tables

**Figure 1 jpm-11-01342-f001:**
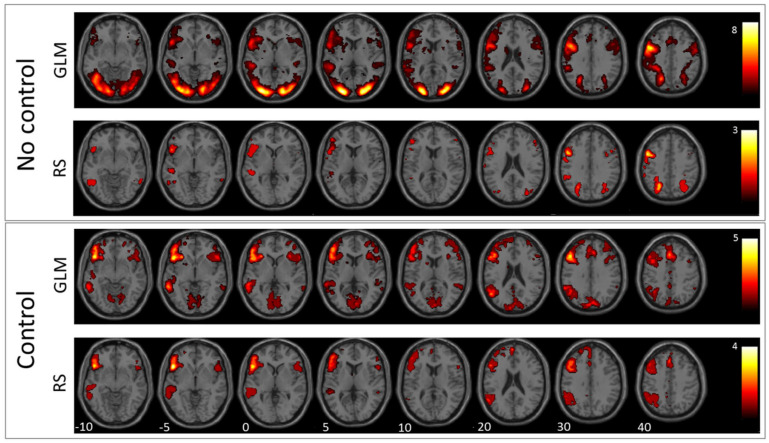
Verb generation: In this figure, the mean images of the used GLM contrast images and selected ICs for the verb generation task are shown. The first row depicts the GLM result without using the control condition and the second row the corresponding RS-fMRI result. The third row shows the mean of the selected language-related ICs when using the GLM contrast with the control condition as a template, and the fourth row shows the same for RS-fMRI.

**Figure 2 jpm-11-01342-f002:**
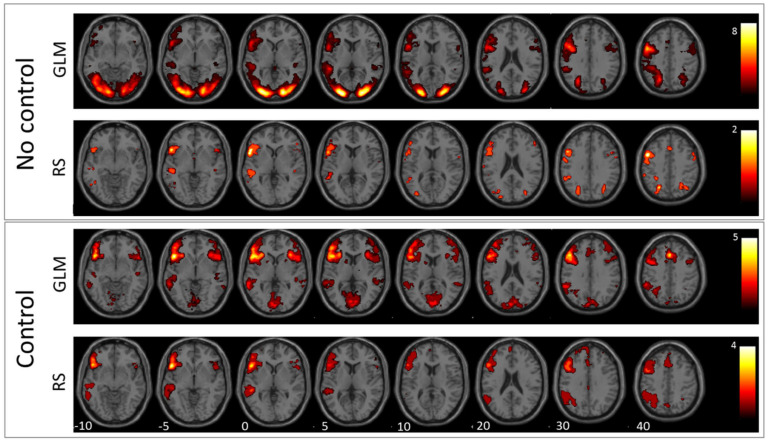
Antonym generation: In this figure, the mean images of the used GLM contrast images and selected ICs for the antonym generation task are shown. The first row depicts the GLM result without using the control condition and the second row the corresponding RS-fMRI result. The third row shows the mean of the selected language-related ICs when using the GLM contrast with control condition as template, and the fourth row shows the same for RS-fMRI.

**Figure 3 jpm-11-01342-f003:**
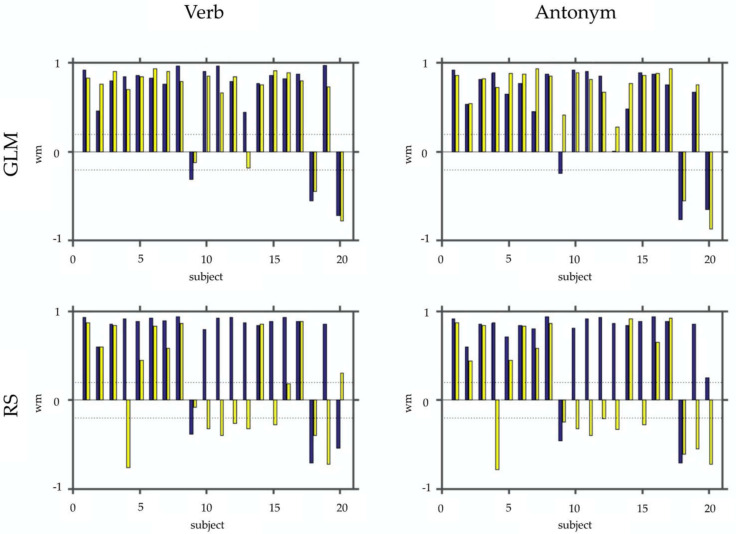
This figure depicts the WMLI on a single subject basis. In the upper row, panels show the results of the SPM contrasts (GLM) for the verb and antonym generation. In the bottom row, the same is shown for the selected RS-fMRI components (RS). Each panel shows bar plots with the WMLI results using a control condition (blue) vs. not using one (yellow). For better interpretation of the WLMI results, we added two dashed lines representing the interval [−0.2; 0.2].

**Figure 4 jpm-11-01342-f004:**
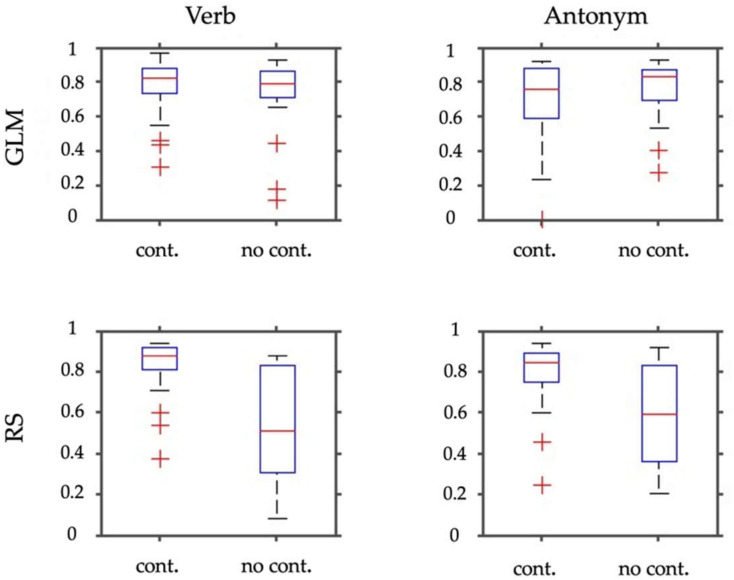
This figure depicts box plots of the WMLI of the laterality index calculations (absolute values). In the upper row, panels show results for the SPM contrasts (GLM) for the verb and antonym generation. The bottom row shows the same for the selected RS-fMRI components (RS). Each panel depicts box plots resulting from employing the task either with the control condition (cont.) or without (no cont.).

**Table 1 jpm-11-01342-t001:** This table describes the parameters of the patient sample in this study. The first column shows the patient number (No.); the second column depicts the tumor localization (Localization) and the third column the side of the lesion (Hemisphere); tumor classification (* no exact classification possible) is described in the fourth column and the type of surgical intervention (GA = craniotomy and tumor removal under general anesthesia, SB = stereotactic biopsy, AAA = craniotomy, and tumor removal using an awake-awake-awake-technique, OB = craniotomy and biopsy (open biopsy)) is described in the fifth column (** no surgery was performed); the Scanner column (number 6) shows whether patients were examined by means of an Allegra (A) or Skyra (S) MRI scanner. The number of scans obtained for each patient in the RS-fMRI session is shown in the seventh column (#Scans RS).

No.	Localization	Hemisphere	Tumor Classification	Surgical Intervention	Scanner	#Scans
						RS
1	frontotemporal	right	Secondary glioblastoma IV	GA	A	311
2	temporal	left	Diffuse astrocytoma II	SB	A	311
3	frontal	left	Anaplastic astrocytoma III	AAA	A	311
4	temporal	right	*	SB	A	311
5	parietal	right	Diffuse astrocytoma II	SB	A	299
6	frontal	left	Anaplastic astrocytoma III	GA	S	300
7	frontal	right	Diffuse astrocytoma II	GA	A	311
8	parietooccipital	left	Secondary glioblastoma IV	GA	S	300
9	frontal	left	Anaplastic astrocytoma III	OB	A	311
10	frontal	left	Oligodendroglioma II	AAA	S	450
11	temporal	left	Glioblastoma IV	GA	A	450
12	temporal	right	Glioblastoma IV	GA	S	450
13	frontal	left	Secondary glioblastoma IV	GA	S	300
14	temporal	left	Meningioma WHO I	GA	S	450
15	temporal	left	Anaplastic astrocytoma III	**	A	311
16	temporal	left	Oligoastrocytoma II	AAA	A	450
17	parietal	left	Arteriovenous malformation	GA	A	311
18	frontotemporal	left	Recurrent glioblastoma IV	GA	A	311
19	frontal	right	Anaplastic astrocytoma III	GA	S	450
20	frontoparietal	right	Recurrent glioblastoma IV	GA	A	311

**Table 2 jpm-11-01342-t002:** This table lists the cluster distributions of each group statistics image. The distributions of clusters are described by the peak coordinates (Peak MNI), the number of voxels in each cluster (#Voxel), the maximum t-value (t_max_), the AAL (automatic anatomic labeling) regions containing at least 10% of the cluster’s voxels, and the percentage of voxels of the cluster in the particular region.

Peak	#Voxel	t_max_	AAL Regions	%	Peak	#Voxel	t_max_	AAL Regions	%
MNI					MNI				
GLM antonym control					GLM verb no control				
−52 28 2	4668	12.4	Left inferior frontal gyrus	63	−46 −2 42	22927	11.1	Left inferior frontal gyrus	13
			Left middle frontal gyrus	17	44 22 −8	441	5.3	Right insula	51
			Left Insula	10				Right inferior frontal gyrus	24
2 −70 6	4520	7.1	Left lingual gyrus	24	48 −32 4	117	4.5	Right superior temporal gyrus	77
			Left calcarine sulcus	18				Right middle temporal gyrus	10
			Left cuneus	12	24 10 6	481	4.7	Right putamen	36
			Right calcarine sulcus	12				Right caudate nucleus	16
34 22 −6	1447	7.2	Right inferior frontal gyrus	59	−28 −50 52	2860	9.2	Left superior parietal lobule	33
−58 −46 0	637	6.1	Left middle temporal gyrus	80				Left inferior parietal lobule	31
48 −26 −6	130	4.6	Right superior temporal gyrus	39				Left postcentral gyrus	13
			Right middle temporal gyrus	28	28 −58 44	1623	9.8	Right superior parietal lobule	30
−4 24 36	1712	7.1	Left supplementary motor area	31				Right inferior parietal lobule	15
			Left medial frontal gyrus	24				Right postcentral gyrus	14
			Right supplementary motor area	13	RS antonym control				
GLM antonym no control					−48 42 −10	2883	5.9	Left inferior frontal gyrus	65
28 −90 8	3472	6.6	Right lobule VI of cerebellar hemisphere	17				Left middle frontal gyrus	18
			Right middle occipital gyrus	15	−106	352	4.9	Left middle temporal gyrus	80
			Right inferior occipital gyrus	14	−14 4 12	111	5.4	Left caudate nucleus	54
			Right fusiform gyrus	13				Left putamen	19
			Right inferior temporal gyrus	11	−50 −54 38	512	6.0	Left inferior parietal lobule	51
−54 −38 6	4753	7.3	Left middle occipital gyrus	24				Left supramarginal gyrus	29
			Left fusiform gyrus	19				Left angular gyrus	19
			Left middle temporal gyrus	15	−2 24 54	674	5.3	Left supplementary motor area	66
			Left inferior occipital gyrus	15				Left medial frontal gyrus	27
−18 0 18	9513	9.8	Left inferior frontal gyrus	30	RS antonym no control				
			Left precentral gyrus	18	−58 6 16	109	4.3	Left inferior frontal gyrus	88
34 24 0	954	6.9	Right inferior frontal gyrus	51				Left precentral gyrus	10
			Right insula	24	−20 −70 34	128	4.4	Left superior occipital	40
−50	204	6.8	Left Putamen	12				Left superior parietal lobule	33
36 −32 2	128	5.4	Right superior temporal gyrus	58				Left middle occipital gyrus	19
18 −6 22	396	6.2	Right caudate nucleus	27	−40 −4 50	141	4.0	Left precentral gyrus	98
			Right pallidum	15	RS verb control				
			Right putamen	12	−46 32 −2	3170	6.2	Left inferior frontal gyrus	63
28 −58 44	851	7.6	Right superior parietal lobule	35				Left middle frontal gyrus	21
			Right inferior parietal lobule	15	−104	292	4.5	Left middle temporal gyrus	86
			Right angular gyrus	13	−16 10 12	194	5.5	Left caudate nucleus	49
−6 8 60	1496	10.8	Left supplementary motor area	55				Left putamen	29
			Right supplementary motor area	26	−50 −54 34	790	5.6	Left inferior parietal lobule	42
28 −4 56	183	4.4	Right superior frontal gyrus	40				Left angular gyrus	32
			Right middle frontal gyrus	31				Left supramarginal gyrus	23
			Right precentral gyrus	11	−2 26 50	818	5.0	Left supplementary motor area	47
GLM verb control								Left medial frontal gyrus	41
−44 22 −2	4253	8.1	Left inferior frontal gyrus	62	RS verb no control				
			Left middle frontal gyrus	24	−20 −66 40	722		Left superior parietal lobule	57
−2 −90 20	2879	6.8	Left cuneus	19				Left middle occipital gyrus	16
			Left calcarine sulcus	17				Left inferior parietal lobule	12
			Left lingual gyrus	16	−40 −2 42	590	5.1	Left precentral gyrus	79
			Right cuneus	13				Left inferior frontal gyrus	11
			Right lingual gyrus	11	32 −70 46	792	5.1	Right superior parietal lobule	44
−82	745	6.3	Left middle temporal gyrus	80				Right inferior parietal lobule	18
34 24 −4	341	5.8	Right insula	50				Right superior occipital gyrus	16
			Right inferior frontal gyrus	36				Right angular gyrus	12
−60 −54 18	200	4.7	Left middle temporal gyrus	53					
			Left supramarginal gyrus	22					
			Left angular gyrus	17					
−4 18 60	1508	6.9	Left medial frontal gyrus	38					
			Left supplementary motor area	30					
			Left anterior cingulum	11					

**Table 3 jpm-11-01342-t003:** Mean and standard deviation of the Dice index correlations of the best fitting ICs to the TB-fMRI results of verb and antonym generation with and without the linguistic control condition (verb = verb generation; anto = antonym generation; w/o LCC = without linguistic control condition; with LCC = with linguistic control condition).

Dice Index Correlations	Verb w/o LCC	Anto w/o LCC	Verb with LCC	Anto with LCC
mean	0.276	0.2825	0.3065	0.2906
standard deviation	0.06102	0.07639	0.10682	0.08484

## Data Availability

The data that support the findings of this study are available from the corresponding author, C.O., upon reasonable request.

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
