# Peer review of "Preoperative Assessment of Language Dominance through Combined Resting-State and Task-Based Functional Magnetic Resonance Imaging"

_jpm, 2021, doi:10.3390/jpm11121342_

Round 1
Reviewer 1 Report
Interesting but unnecessarily verbose.
I recommend they shorten the text.
The paper is relevant, well-conducted and well-written.
Author Response
Response to Reviewer 1 Comments:
Dear Reviewer,
we thank you for the opportunity to revise our manuscript entitled “Preoperative assessment of language dominance through combined resting-state and task-based functional magnetic resonance imaging.”
We are very grateful for your helpful suggestions and comments. We have modified the manuscript accordingly. Detailed responses comments are listed below.
With best regards
Christian Ott
- Interesting but unnecessarily verbose. I recommend they shorten the text.
Thank you very much for your considerations. We see that especially the Introduction and the Discussion are a bit lengthy. For this reason, we shortened the text by excluding some passages (for instance, the fundamental principles of TB-fMRI and RS-fMRI). We also tried to preserve comprehensibility. Please see the revised manuscript for details.

Reviewer 2 Report
The manuscript uses RS-MRI and TB-MRI in patients with brain lesions to determine brain dominance. There are other examples in the literature looking at RS-MRI in patients with lesions, therefore the novelty of the work.
Major Revisions:
1) The authors should consider making the introduction to be more concise. There are too many separate short paragraphs and no transitions between ideas. The ideas seem to follow 2-3 themes these should be the main paragraphs and then the author should complete the intro with a paragraph describing the work of the paper. The discussion suffers from the same issues.
2) The authors should improve the quality of their figures, particularly for Figures 3 & 4.
Minor Revisions:
1) rs-fMRI, Rs-fMRI, and RS-fMRI are used across the manuscript, the authors should select one and be consistent.
2) Figure 1 and 2 are very similar, it would be helpful to title them more clearly with the task type upfront, so that the readers do not need to search.
3) In the first paragraph of page 10, "(for details see Figure 1, Tab. 2)" is the only place table is not written out in full.
4) The statements "Authors should discuss the results and how they can be interpreted from the perspective of previous studies and of the working hypotheses. The findings and their implications should be discussed in the broadest context possible. Future research directions may also be highlighted." and "This section is not mandatory but can be added to the manuscript if the discussion is unusually long or complex" were accidentally left at the end of the discussion and conclusions.
Author Response
Response to Reviewer 2 Comments:
Dear Reviewer,
we thank you for the opportunity to revise our manuscript entitled “Preoperative assessment of language dominance through combined resting-state and task-based functional magnetic resonance imaging.”
We are very grateful for your helpful suggestions and comments. We have modified the manuscript accordingly. Detailed responses comments are listed below.
With best regards
Christian Ott
- The authors should consider making the introduction to be more concise. There are too many separate short paragraphs and no transitions between ideas. The ideas seem to follow 2-3 themes these should be the main paragraphs and then the author should complete the intro with a paragraph describing the work of the paper. The discussion suffers from the same issues.
Thank you very much for your considerations. We see that especially the Introduction and the Discussion are a bit lengthy. For this reason, we shortened the text by excluding some passages (for instance, the fundamental principles of TB-fMRI and RS-fMRI). We also tried to preserve comprehensibility. We changed the paragraphs according to the subject matter, and we revised the section describing the work of the paper. Please see the revised manuscript for details.
- The authors should improve the quality of their figures, particularly for Figures 3 & 4.
We see your concern, especially for the annotations, and changed the figures in the revised manuscript accordingly.
Minor Revisions:
- rs-fMRI, Rs-fMRI, and RS-fMRI are used across the manuscript, the authors should select one and be consistent.
Thank you very much for your suggestion. We changed it throughout the manuscript to RS-fMRI and TB-fMRI.
- Figure 1 and 2 are very similar, it would be helpful to title them more clearly with the task type upfront, so that the readers do not need to search.
We added the task in brackets directly after the figure number.
- In the first paragraph of page 10, "(for details see Figure 1, Tab. 2)" is the only place table is not written out in full.
We changed this in the revised manuscript.
- The statements "Authors should discuss the results and how they can be interpreted from the perspective of previous studies and of the working hypotheses. The findings and their implications should be discussed in the broadest context possible. Future research directions may also be highlighted." and "This section is not mandatory but can be added to the manuscript if the discussion is unusually long or complex" were accidentally left at the end of the discussion and conclusions.
We apologize for this and changed the manuscript accordingly.

Round 2
Reviewer 1 Report
Acceptable revision